# Pre-Birth Household Challenges Predict Future Child’s School Readiness and Academic Achievement

**DOI:** 10.3390/children9030414

**Published:** 2022-03-15

**Authors:** Robyn A. Husa, Jared W. Parrish, Heidi S. Johnson

**Affiliations:** 1Alaska Mental Health Board, Alaska Department of Health and Social Services, Juneau, AK 99811, USA; robyn.husa@alaska.gov; 2Section of Women’s, Children’s and Family Health, Division of Public Health, Alaska Department of Health and Social Services, Anchorage, AK 99503, USA; 3Independent Researcher, Douglas, AK 99824, USA; heidijohnsonslp@gmail.com

**Keywords:** household challenges, ACEs, pre-birth, early development, reading, school readiness, PRAMS

## Abstract

Early developmental success and school readiness strongly influence future skill development, occupational opportunities, and health. Therefore, it is critical to identify and address early determinants of school readiness for supporting children’s overall well-being and success. In this retrospective cohort study, we examined the effects of pre-birth household challenges, such as homelessness or experiences of intimate partner violence, on children’s early school readiness. We linked data from the Alaska 2009–2011 Pregnancy Risk Assessment Monitoring System (PRAMS) to administrative and education records through 2019. Education records included kindergarten developmental scores, third grade reading assessments, and attendance records. Generalized linear models with Quasi-Poisson distributions for each outcome of interest examined the predictive value of pre-birth household challenges on the risks of not meeting school readiness expectations. We found that experiencing higher numbers of pre-birth household challenges was related to higher risk of the child not meeting developmental and reading proficiency and having chronic absenteeism. These results suggest that it is imperative support systems for pregnant persons and their families be introduced as soon as possible in pre-natal care routines to address current pre-birth household stressors and prevent future challenges. Such early prevention efforts are needed to ensure the best possible developmental start for children.

## 1. Introduction

The pre-natal period, including the health and well-being of the pregnant parent, is crucial for a child’s early developmental success [1]. High pre-natal stress experienced by the birthing parent is associated with suboptimal cognitive and socio-emotional outcomes for school aged children (for reviews, see [2,3]). Such outcomes underly a child’s school readiness, or how prepared they are to succeed in school cognitively, socially, physically, and emotionally [4]. Early developmental success and school readiness influence future skill development, occupational opportunities, and health [5,6,7]. Therefore, identifying and addressing early determinants of school readiness is important for supporting children’s overall well-being and success. As such, major stressful pre-birth challenges experienced by the birthing parent around the time of pregnancy may have profound negative impacts on their child’s school readiness and future achievement. The current study aimed to clarify this relationship between pre-birth household challenges experienced by the birthing parent and their potential impact on the child’s school readiness and achievement in a longitudinal Alaskan cohort.

Before children enter kindergarten, they have already experienced a plethora of early educational experiences that shape their cognitive and socio-emotional development [8]. The ongoing interaction between a child and learning opportunities within their environment influences the child’s level of readiness to learn upon entrance into school (i.e., school readiness). This presenting “readiness” further interacts with a school’s ability to receive and support the various abilities of children, which also interacts with the underlying family’s and community’s ability to support continued optimal early child development [9]. Readiness to learn typically includes a child’s physical well-being and sensory motor development, social and emotional development, approaches to learning (e.g., enthusiasm, temperament), language development, and general knowledge and cognition (e.g., literacy and math skills) [4,10]. Higher levels of school readiness consistently predict higher later academic achievement [11,12,13], which in turn has been linked to multiple improved economic and health outcomes [5,6,7]. The focus of the current study was to identify early determinants of a child’s personal school readiness and early academic achievement, key pieces in the chain of influences to future educational success.

Prior research demonstrates that both mutable and immutable pre-birth factors are linked to early childhood outcomes. In a predictive risk model for school readiness developed by Camacho and colleagues [14], the most important pre-natal or at birth variables found in predicting school readiness were related to socioeconomic conditions (i.e., social class, maternal education, and family income) and the child’s ethnicity. This model falls in line with prior studies identifying socioeconomic determinants of school readiness, such as maternal education, economic disadvantage, and single-parent family status [15,16]. Treatable parental health factors such as pre-natal substance use [17,18,19,20,21] and mental health concerns experienced by the birthing parent [22,23] (for reviews, see [2,24]) also increase the risk of the child not meeting developmental standards needed for proper school readiness. It should be noted that certain substance uses during pregnancy, such as smoking, may be a consequence of or may interact with deeper familial socioeconomic and psychopathological risk factors, rather being than a causal determinant of school readiness in and of itself [25]. Overall, multiple mutable pre-birth challenges and risk factors interact with demographic factors to predict risk for children not meeting school readiness ideals by the time they enter the education system at kindergarten.

Additional research has connected the accumulation of the above and other types of pre-birth stressors to school readiness. Children born to birthing parents who self-reported experiencing higher levels of pre-natal stress through natural disaster exposure (e.g., loss of personal and/or business income, exposure to physical dangers, change in family dynamic or place of residence) demonstrated lower general intellectual and language abilities at age 2 [26], prior to entering kindergarten. One study documented that exposure to four or more pre-birth stressors, such as job loss and marital issues, was negatively associated with literacy scores at an even later developmental stage (age 10) for female children [27]. Potentially, experiencing such high levels of stress during pregnancy could make it difficult for the family to provide the extra needed support for optimal early development to their future child due to a focus on current basic needs.

While prior research has identified risk factors for not meeting early school readiness related to perinatal stressors, no study to these authors’ knowledge has directly studied the effect of pre-birth household challenges experienced by the birthing parent around pregnancy on multiple measures of school readiness and early academic achievement of the child in a representative statewide longitudinal cohort. To fill this knowledge need, the current study examined whether the number of pre-birth household challenges experienced by the birthing parent predicted the child’s school readiness and early academic achievement, as measured by performance on the Alaska kindergarten developmental profile assessment, third grade reading assessments, and average school attendance.

## 2. Materials and Methods

### 2.1. Data Sources

This project used retrospective data from the Alaska 2009–2011 Pregnancy Risk Assessment Monitoring System (PRAMS). Alaska PRAMS is a population-based weighted sample of birthing parents delivering live births in Alaska. Birthing parents are surveyed about factors related to pre-pregnancy, pregnancy, and post-birth experiences. The 2009–2011 PRAMS phase sampled 5578 of the 33,709 eligible Alaska resident births, with oversampling on birthing parents who identified as Alaska Native and who had newborns with low birthweight status (<2500 g). The average weighted response weight was 66%. The complete PRAMS survey methodology is described elsewhere [28].

Leveraging the Alaska Longitudinal Child Abuse and Neglect Linkage Project (ALCANLink), which links PRAMS survey responses with multiple administrative records (see [29,30] for description of sources linked), we integrated PRAMS survey responses with Department of Education and Early Development (DEED) records through 2019. DEED records included the cohort children’s 3rd grade Performance Evaluation for Alaska’s Schools (PEAKS) English language arts assessment scores, Alaska Developmental Profile (ADP) scores, and attendance records. PEAKS English language arts (ELA) evaluation assesses students’ skills in reading complex texts, writing with clarity, and presenting and evaluating ideas. Students can receive a PEAKS ELA score of Advanced, Proficient, Below Proficient, or Far Below Proficient. The ADP is an Alaskan developed evaluation tool given to students entering kindergarten or first grade which identifies whether students are consistently demonstrating 13 goals and indicators in the following five domains from Alaska’s Early Learning Guidelines: (1) physical well-being, health, and motor development; (2) social and emotional development; (3) approaches to learning; (4) cognition and general knowledge; (5) communication, language, and literacy. Students can receive a score range of 0–13 goals met, and meeting at least 11 out of 13 goals overall is considered to be the developmental gold standard in this assessment (DEED, https://education.alaska.gov/assessments/developmental, accessed on 14 March 2022).

### 2.2. Measures

#### 2.2.1. Exposures

The 2009–2011 PRAMS cohort (*n* = 3549 respondents) data were used to identify household challenges experienced by the birthing parent during the pre-birth period (typically during the 12 months before birth). Household challenges were identified from Centers for Disease Control and Prevention (CDC) standardized PRAMS questions regarding stressful life events [20]. Self-reported pre-birth household challenges included: having a close family member become very sick and go into the hospital, experiencing a divorce or separation, moving to a new address, experiencing homelessness, the birthing parent losing their job, the partner or spouse losing their job, arguing with a partner or spouse more than usual, having a lot of bills that could not be paid, being in a physical fight, the birthing parent or partner or spouse being jailed, someone close to the birthing parent having a problem with drugs or drinking, experiencing a death of someone close to the birthing parent, experiencing intimate partner violence, and the birthing parent being checked or treated for anxiety or depression by a medical professional.

The total number of household challenges experienced was calculated by adding up the number of components respondents endorsed with a “yes”. Missing responses were treated as a 0 or “no” response when calculating the total household challenges. Due to small sample size, we then categorized the number of pre-birth household challenge components experienced into a single construct with the following categories: 0–1 reported, 2–3 reported, and ≥4 reported. Categories were developed based on prior research of household-challenge-based risk groups [23]. Respondents who did not answer any of the exposure questions were excluded from the analyses.

#### 2.2.2. Outcomes

The first outcome of interest was the child’s kindergarten developmental profile (ADP score). We created a bivariate (≥11 goals met, <11 goals met) ADP variable consistent with DEED’s developmental gold standard definition. The second outcome of interest was the child’s 3rd grade PEAKS ELA score. We created a bivariate PEAKS variable by categorizing Advanced and Proficient (A/P) scores together as “met proficiency” and Below Proficient and Far Below Proficient (BP/FBP) as “did not meet proficiency.” The final outcome of interest was chronic absenteeism, defined as missing at least 10% of the days in which a student was enrolled in school. The child’s school attendance records were averaged across the 2015–2019 school years, and we created a bivariate variable from the resulting average to indicate the presence of chronic absenteeism (≥90% attendance, <90% attendance).

#### 2.2.3. Covariates

Prior research working with Alaska population data [31,32] identified demographic variables significantly associated with negative early childhood outcomes: Alaska Native/American Indian race status, lower maternal education, and unmarried maternal marital status. Maternal education status has been associated with school readiness outcomes in non-Alaska data as well [14]. Therefore, Alaska Native race status, maternal education, and maternal age at the time of the birth were included as *a priori* covariates in multivariate analyses. It is important to note that the above demographic variables do not represent causal factors or biological predispositions toward certain early childhood outcomes, but likely represent a population experiencing differential distribution of underlying modifiable risk factors (e.g., systemic challenges, lack of additional social or economic supports).

### 2.3. Statistical Analysis

Using the PRAMS post-stratification population weights, we derived the estimated proportion of the birth population reflective of each component of the demographic, exposures, and outcome classifications.

We separately examined the bivariate association of each individual pre-birth household challenge component and risk of not meeting PEAKS proficiency, not meeting the 11-goal threshold on ADP, and being chronically absent by calculating risk ratios with “no exposure” as the reference category. We then constructed two generalized linear models with Quasi-Poisson distributions for each outcome of interest to examine the predictive value of pre-birth household challenges on the risks of not meeting school readiness expectations. We used Quasi-Poisson models to account for overdispersion, which relaxes the assumption that the variance is equal to the mean and instead assumes the variance is a linear function of the mean. First, we separately modeled the association of the total number of pre-birth household challenges (categorized as 0–1, 2–3, and 4+ challenges) on ADP scores, PEAKS scores, and attendance records. Outcome reference categories were A/P (PEAKS), ≥11 goal met (ADP), and ≥90% attendance (attendance), respectively. Secondly, we constructed a multivariate model adjusting for Alaska Native race (Native, non-Native), maternal education (≥12 years, <12 years), and maternal age (≥20 years, <20 years) at the time of the birth to understand the individual association of each component. We used backward elimination stepwise regression to remove nonsignificant covariates to establish our final most parsimonious prediction models. Covariates were retained in the final model if their removal produced a >10% change in the effect estimate.

All analyses were conducted using R version 4.1.0 and either the survey [33] or srvyr [34] package.

### 2.4. Institutional Review Board

The PRAMS, DEED, and administrative records used in this study were examined retrospectively under routine public health surveillance. Full details on the PRAMS institutional review board (IRB) approval are found in the Institutional Review Board Statement subsection at the end of this article. The remainder of the current study involved the linkage of existing legally authorized administrative databases housed within the Alaska Department of Health and Social Services (DHSS) and DEED. Under these circumstances, IRB approval was not required or sought for the current study.

## 3. Results

### 3.1. Participant Characteristics

A total of 3549 birthing parents responded to the PRAMS survey, which represents 33,417 (±233) children born in Alaska during 2009–2011. Table 1 presents demographic characteristics of the respondents. The distribution of the number of household challenges within the population is presented in Table 2.

### 3.2. Pre-Birth Household Challenges and Developmental Profile Score

Overall, 69% of the population fell below the developmental profile gold standard (<11 goals met; Appendix A presents the pre-birth household challenge components’ unadjusted association with developmental profile score. Experiencing homelessness was associated with the highest risk of not meeting ADP gold standards (Risk Ratio (RR) = 1.23, 95% Confidence Interval (CI) = 1.09, 1.38), followed by experiencing a divorce or separation (RR = 1.16, 95% CI = 1.05, 1.28). Children born to mothers who reported experiencing ≥4 pre-birth household stressors had 1.16 times the risk of not meeting developmental profile standards than those who were born to mothers who reported experiencing 0–1 household stressors, after adjusting for birthing parent Alaska Native race and years of education at the time of the birth (Table 3). Experiencing 2–3 pre-birth household challenges did not significantly predict risk of not meeting the standard developmental goals.

### 3.3. Pre-Birth Household Challenges and Third Grade Reading Score

Overall, 64% of the population did not meet proficiency in PEAKS third grade reading assessment. Appendix A presents the pre-birth household challenge components’ unadjusted association with third grade reading score. The presence of any pre-birth household challenge, except the birthing parent losing their job or having a sick family member in the hospital, was significantly associated with increased risk of the child not meeting PEAKS third grade reading score proficiency.

The number of reported pre-birth household challenges predicted third grade readings scores in a stepwise, dose–response manner (Table 3). Children born to mothers who reported experiencing ≥4 and 2–3 household stressors during the 12 months before birth had 1.36 and 1.27 times the risk of not meeting reading proficiency, respectively, than those who were born to mothers who reported experiencing 0–1 household stressors, after adjusting for birthing parent Alaska Native race and years of education at the time of the birth.

### 3.4. Pre-Birth Household Challenges and School Attendance

Within the population of interest, 22% met the definition for chronic absenteeism (school attendance <90%). Chronic absenteeism was significantly associated with the birthing parent facing homelessness, being jailed or their partner being jailed, divorce or separation, death of a close friend or family member, increased frequency of arguments with partner or spouse, or drug or alcohol abuse by someone close to the birthing parent during the pre-birth period (Appendix A). Only children born to birthing parents who reported experiencing ≥4 pre-birth household stressors had a significantly higher risk of chronic absenteeism than those who were born to mothers who reported experiencing 0–1 household stressors, after adjusting for birthing parent Alaska Native race and years of education at the time of the birth (Table 3).

## 4. Discussion

This study investigated the relationships between pre-birth household challenges and children’s school readiness and academic achievement within an Alaska statewide representative birth cohort. Indicators of school readiness and achievement included performance on kindergarten developmental profiles, third grade reading evaluation scores, and average school attendance across five years. We discovered that children born into households that experienced high levels of pre-birth household challenges (4+ challenges) were at an increased risk of not meeting future kindergarten developmental goals and experiencing chronic absenteeism compared to children born in households that experienced one or no pre-birth challenges. The level of pre-birth household challenges experienced also had a dose–response relationship with the child’s third grade reading evaluation scores, with higher levels of pre-birth household challenges (2–3 and 4+ challenges) being associated with a stepwise increase in the risk of the child not meeting reading proficiency compared to children born in households who experienced 1 or no pre-birth challenges. Considering that early developmental success and school readiness influences future achievement, these results suggest that it is imperative support systems for pregnant persons and their families are introduced as soon as possible in the normal pre-natal care routine to address current pre-birth household stressors and prevent future challenges. Such early prevention efforts are needed to ensure the best possible developmental start for children.

Results from this study align with and expand upon prior research highlighting links between individual pre-natal stressor exposure and negative education outcomes of offspring. Consistent with prior research linking economic disadvantage during the pre-natal period with the future children not meeting school readiness expectations [14,15,16], in our study, we documented that heavy financial burden, such as homelessness, job loss, and divorce, were consistently independently associated with greater negative impacts on school readiness. Furthermore, when taking each challenge together, our results strongly suggest an additive effect, and the most noticeable school readiness consequences can be seen at the highest levels of pre-natal household challenge experiences. The prevailing rationale for the additive influence of stressors on child development suggests that combined financial and other pre-birth stressors may simply overwhelm parents and lead them to be focused on survival and basic needs (e.g., paying rent, putting food on the table), making it difficult to provide the optimized physical and psychological home environment supportive of learning and development for the child [35]. Finally, a strength of this study was that it used multiple measures and timepoints to examine these pre-birth household challenge effects on school readiness and achievement rather than relying on a single snapshot of early education abilities. In this way, we were able to not only show pre-birth household-challenge-related consequences on early, pre-grade school readiness outcomes, but also continuous negative impacts within the same cohort in later education achievement and engagement.

There are several potential reasons why a parent’s experiences of stress or challenges prior to the birth of their child would affect the child’s eventual school readiness and early academic achievement. Parents experiencing financial-based challenges, such as homelessness or losing their job(s), may not be able to afford basic pre-natal care or, if the challenge continues post-birth, to enroll their child in beneficial early success programs such as preschool. Poor or absent pre-natal care can negatively affect the child’s physical development (e.g., low birth weight), which in turn can have negative effects on school readiness [36]. In addition, parents may not be able to afford to purchase quality children’s books or to spend long periods of time reading to their child (as opposed to searching for work or housing), both of which are influential to a child’s early reading skills and development [37,38]. Furthermore, experiencing higher pre-birth household challenges predicts an increased risk of the child having higher adverse childhood experiences by the age of 3 years [31], and adverse childhood experiences have long been linked to poor developmental, health, and educational outcomes (e.g., for review, see [39]), including reduced school readiness [40]. Finally, the parents’ stress may limit their awareness of, and thus access to, beneficial child development programs such as Head Start and other early childhood educational opportunities. Therefore, pre-birth household challenges that the birthing parent experiences can lead to a chain reaction of maladaptive events or experiences that eventually disadvantage the child’s school readiness.

Person- or family-level prevention efforts and programs may mitigate the effects of pre-birth household challenges on children’s school readiness and early learning development. In the pre-birth window, primary care physicians and related health care providers can screen pregnant individuals for the presence of these household challenges. If significant challenges are present, health care providers can guide the pregnant individual and their family toward public health resources and care coordinators aimed at helping families navigate and connect with financial, social, or other supports services needed. Comprehensive programs that address family-specific challenges are effective. Children whose families participated in a nurse home visiting program from the pre-natal period through the child’s infancy demonstrated higher intellectual functioning and fewer clinical behavioral issues [41]. Other interventions can be engaged in at the perinatal and early childhood periods to mitigate the transmission of stress and subsequent risk of poor outcomes from the parent to the child. One example of this is the Child FIRST program [42], which identifies children in high-risk families as early as possible and provides in-home assistance geared toward decreasing psychosocial stress, promoting connection to integrated services and supports, and promoting responsive, nurturing caregiving through a relationship-based psychotherapeutic approach. Children in families enrolled in Child FIRST had improved language development compared to those who experienced usual care, and birthing parents in the program had less parenting stress and protective service involvement compared to those following usual care. Evidence-based pre-natal and perinatal programs that not only address pre-birth household challenges, but work to mitigate their effects on the child and birthing parent post-birth should be readily available and easily accessible for any pregnant individual and their family in order to curb risks of the child not meeting early educational development markers.

There are some limitations to the current study. One limitation is that the study was conducted using an Alaska-based population, making the population demographic distribution different from the general United States population. However, this study can serve as a generalizable platform for other states and areas to adopt when examining pre-birth effects on children’s school readiness. Another limitation is that PRAMS responses are self-reported and may reflect social-desirability bias toward not reporting on sensitive experiences. However, results from the PRAMS survey show 19% of our cohort reported high levels of household challenges (4+), which is comparable to the U.S. percentage of adults who reported feeling high levels of stress over the years during which PRAMS was conducted (24% in 2009 and 2010, 22% in 2011; www.stressinamerica.org, accessed on 19 February 2022). However, it should be noted there are individual differences in subjective feelings of stress following challenging experiences. This study is also limited by the lack of potential heterogeneity in education outcomes, with over 50% of Alaskan children not meeting the developmental profile standard or third grade reading score proficiency levels. Additional research that accounts for the pre-birth and early childhood experiences are critical, especially considering the length of time between the stressors being experienced in this study and the school outcomes being measured. However, the current study indicates that despite this limitation, initiating and supporting families in some key areas may have impacts on school readiness of offspring years later. Finally, there is a limitation in the calculation of the household challenge total scores used in the bivariate analyses. In constructing these scores, any missing response was counted as a “no” response, which could lead to underestimated counts.

Future research should focus on ways in which public health can target and mitigate the effects of pre-birth household stressors on children’s school readiness and development and the impact of care coordination to assist families dealing with multiple challenges. Several intervention strategies were mentioned above, but none specifically addressed the relationship between pre-birth household challenges and targeted comprehensive school readiness measures as outcomes of interest. It is important to note that the relationship between pre-birth household stressors and school readiness is likely mediated by several factors, such as parent−child attachment, that were beyond the scope of the current project. These factors and intervention efforts to target them can be addressed by future public health and education researchers.

Education readiness and early school performance are strong predictors of future education success and graduation. Education success and graduation are strong predictors of health and economic attainment. Understanding these relationships is critical for developing a cross-sector approach including education, public health, and medical professionals. Thus, while efforts are made to restructure and improve the educational system, we also need to focus on supporting families early on and to recognize them as the first “educational institution” that establishes the learning foundation. This would increase the number of children entering school ready to learn.

## 5. Conclusions

The current study examined whether the number of pre-birth household challenges experienced by the birthing parent predicted the child’s early school readiness, as measured by kindergarten developmental profiles, third grade reading assessments, and average school attendance. We found that experiencing higher numbers of pre-birth household challenges was related to higher risk of the child not meeting developmental and reading proficiency and having chronic absenteeism. Experiencing homelessness was consistently the highest risk pre-birth challenge for poor school readiness outcomes. These results demonstrate how important the pre-birth window is for providing familial support programs, particularly financial, early on in an individual’s pregnancy to start children on a path to educational success immediately at birth.

## Figures and Tables

**Table 1 children-09-00414-t001:** Demographic characteristics of PRAMS respondents (birthing parents) and offspring.

Variable	*N*	Weighted Mean (95% CI)
**Sex of Child**		
Male	1822	0.52 (0.49, 0.54)
Female	1727	0.49 (0.47, 0.51)
**County Type**		
Urban	2843	0.87 (0.86, 0.89)
Rural	477	0.13 (0.11, 0.14)
Missing	7	-
**Marital Status**		
Married	2067	0.62 (0.60, 0.64)
Not Married	1477	0.38 (0.36, 0.40)
Missing	5	-
**Level of Education**		
≥12 Years	2843	0.87 (0.86, 0.89)
<12 Years	477	0.13 (0.11, 0.14)
Missing	229	-
**Race**		
Alaska Native/Native American	1257	0.26 (0.25, 0.26)
Non-White/Non-Native	387	0.12 (0.11, 0.14)
White	1715	0.62 (0.61, 0.64)
Missing	190	-
**Medicaid**		
Yes	2053	0.53 (0.51, 0.55)
No	1496	0.47 (0.45, 0.49)

PRAMS: Pregnancy Risk Assessment Monitoring System. Medicaid: Whether Medicaid covered birthing expenses.

**Table 2 children-09-00414-t002:** Distribution of number of household challenges reported by PRAMS survey respondents.

Stressor Count	*N*	Weighted Mean (95% CI)
4+	680	0.19 (0.17, 0.21)
2–3	955	0.26 (0.24, 0.27)
0–1	1914	0.56 (0.54, 0.58)

PRAMS: Pregnancy Risk Assessment Monitoring System.

**Table 3 children-09-00414-t003:** Relative comparison of expected kindergarten developmental profile score, third grade reading proficiency score, and average school attendance rate by number of pre-birth household challenges.

Number of Pre-Birth Household Challenges	PRAMS Respondents (*n* = 3549)		Risk Ratio (95% CI)	Adjusted Risk Ratio (95% CI) ^a^
ADP	≥11 Goals *	<11 Goals		
4+	106	358	1.15 (1.07, 1.24)	1.16 (1.07, 1.25)
2–3	176	465	1.03 (0.95, 1.12)	1.02 (0.94, 1.11)
0–1	404	916	Referent	Referent
PEAKS	A/P *	BP/FBP		
4+	53	208	1.40 (1.24, 1.57)	1.36 (1.21, 1.53)
2–3	94	253	1.29 (1.15, 1.45)	1.27 (1.14, 1.42)
0–1	252	424	Referent	Referent
Attendance	≥0.90% *	<0.90%		
4+	369	148	1.38 (1.12, 1.70)	1.29 (1.04, 1.60)
2–3	515	203	1.20 (0.99, 1.46)	1.12 (0.93, 1.36)
0–1	1105	351	Referent	Referent

^a^ Education outcome scores adjusted for birthing parent Alaska Native race and years of education at time of the birth. * Referent category for outcome variable. PRAMS: Pregnancy Risk Assessment Monitoring System. ADP: Alaska Developmental Profile. PEAKS: Performance Evaluation for Alaska’s Schools Reading Assessment. A/P: Score of Advanced or Proficient on PEAKS. BP/FBP: Score of Below Proficient or Far Below Proficient on PEAKS.

## Data Availability

The data presented in this study are available as a restricted access limited data set on request to the corresponding author and with approval by DEED administrators. Due to privacy and legal constraints, access to these data requires a data use agreement and research and protocol review.

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
