# Peer review of "Pre-Birth Household Challenges Predict Future Child’s School Readiness and Academic Achievement"

_children, 2022, doi:10.3390/children9030414_

Round 1

Reviewer 1 Report

This is a very well written and presented article which will be of interest to readers. 

Author Response

Thank you for your time reviewing our manuscript and for your positive comments. We are glad that you believe this research will be of interest to readers.

Reviewer 2 Report

The manuscript was well written

Author Response

Thank you for your time reviewing our manuscript and for your positive comments.

Reviewer 3 Report

Thank you for the opportunity to review this manuscript assessing the link between prenatal risk factors and school readiness in a sample from Alaska.  Overall, I found it to be a rigorous study and a well-written manuscript.  My main suggestions for improvement are in strengthening the background section and re-working the discussion.

Introduction:

  1.  The background section is pretty shallow.  The authors make broad statements about how prenatal risk factors can affect school readiness, but do not go in depth into individual factors.  After reading the introduction, it wasn't clear to me what specific factors they would be examining and their theoretical link to school readiness.
  2. More information on the outcome is needed as well.  What is school readiness, why does it matter, etc.
  3. A theoretical framework to support their specific selection of risk factors is needed.

Methods:

  1.  The methods section is very thorough and well written.  I wish the level of specificity in this section had been carried throughout the background section.  
  2. The authors should explain their choice for a quasi-poisson distribution.  Why is that needed in this model?  
  3. The authors talk about control variables in the analysis section, but that should be included in the measures section, with justification for the selected variables.

Results: 

  1. The tables and written results are clear and easy to understand.

Discussion:

  1. In this section, the authors describe why different risk factors might be related to school readiness.  This is the type of information I was looking for in the introduction section.  I would recommend moving this around, and then using the discussion section to focus on how your findings fit in with the prior work, how they extend the findings, and what the implications are.

Author Response

Thank you very much for your thorough review of our manuscript and your constructive feedback. Overall, to address your thoughtful comments, we have strengthened the background section, added more detail in the methods, added references, and modified the discussion. Specifically, we did the following to address your comments (new line references are for tracked changes document):

  • In the Introduction, we have added a paragraph (new lines 45-59) explaining the definition of school readiness more in detail and why it is such an important factor to focus research and intervention efforts toward.
  • To flesh out our theoretical framework, we have added additional language to the Introduction (new lines 77-87), some of which was moved from the Discussion section, as you suggested.
  • A new sub-section (2.2.3.) called “Covariates” (new lines 163-174) was added to the Materials and Methods section. In this sub-section, we detail which covariates were chosen and the reasoning behind their inclusion.
  • A sentence was added to the 2.3 Statistical Analysis sub-section to explain the choice behind a Quasi-Poisson model (new lines 185-187).
  • The discussion section was expanded to include a paragraph (new lines 291-310) stating how our results were in-line with and expanded upon prior research.
  • The paragraph explaining potential reasons why pre-birth stressors could affect the future child’s school readiness (lines 311-333) was kept in the Discussion section rather than moved to the Introduction, as suggested, because we felt it provided more depth as post-hoc explanation of results, with the acknowledgment that there are likely many other underlying reasons (including neurological development within the womb) that were beyond the scope of this article. However, the paragraph was shortened as elements relevant to the theoretical framework of the paper were moved up into the Introduction section.